# Aspergillus Sensitization and Allergic Bronchopulmonary Aspergillosis in Asthmatic Children: A Systematic Review and Meta-Analysis

**DOI:** 10.3390/diagnostics13050922

**Published:** 2023-03-01

**Authors:** Ritesh Agarwal, Valliappan Muthu, Inderpaul Singh Sehgal, Sahajal Dhooria, Kuruswamy Thurai Prasad, Kathirvel Soundappan, Shivaprakash Mandya Rudramurthy, Ashutosh Nath Aggarwal, Arunaloke Chakrabarti

**Affiliations:** 1Department of Pulmonary Medicine, Postgraduate Institute of Medical Education and Research (PGIMER), Chandigarh 160012, India; 2Department of Community Medicine and School of Public Health, Postgraduate Institute of Medical Education and Research (PGIMER), Chandigarh 160012, India; 3Department of Medical Microbiology, Postgraduate Institute of Medical Education and Research (PGIMER), Chandigarh 160012, India; 4Doodhadhari Burfani Hospital, Haridwar 249411, India

**Keywords:** fungal asthma, allergic bronchopulmonary mycosis, ABPM, cystic fibrosis, fungal sensitization, allergic fungal airway disease

## Abstract

**Background**: The prevalence of aspergillus sensitization (AS) and allergic bronchopulmonary aspergillosis (ABPA) in asthmatic children remains unclear. **Objective**: To systematically review the literature to estimate the prevalence of AS and ABPA in children with bronchial asthma. **Methods**: We searched the PubMed and Embase databases for studies reporting the prevalence of AS or ABPA in pediatric asthma. The primary outcome was to assess the prevalence of AS, while the secondary outcome was to evaluate the prevalence of ABPA. We pooled the prevalence estimates using a random effects model. We also calculated the heterogeneity and publication bias. **Results**: Of the 11,695 records retrieved, 16 studies with 2468 asthmatic children met the inclusion criteria. Most studies were published from tertiary centers. The pooled prevalence of AS in asthma (15 studies; 2361 subjects) was 16.1% (95% confidence intervals [CI], 9.3–24.3). The prevalence of AS was significantly higher in prospective studies, studies from India, and those from developing countries. The pooled prevalence of ABPA in asthma (5 studies; 505 children) was 9.9% (95% CI, 0.81–27.6). There was significant heterogeneity and publication bias for both outcomes. **Conclusions**: We found a high prevalence of AS and ABPA in asthmatic children. There is a need for community-based studies from different ethnicities using a standard methodology to ascertain the true prevalence of AS and ABPA in pediatric asthma.

## 1. Introduction

Asthma is the most common chronic respiratory ailment affecting children [1] and is among the top 20 diseases associated with years lived with disability [2]. In the International Study of Asthma and Allergies in Childhood, the prevalence of asthma in children aged 13–14 years ranged from 1.6 to 28.2% (overall, 14.1%), while the prevalence in those 6–7 years was 1.4 to 27.2% (overall, 11.7%) [3,4]. About 1–5% of children can develop severe asthma that is uncontrolled despite high doses of inhaled steroids [5,6]. Several factors are associated with severe asthma, including a family history of asthma, exposure to secondhand smoke, and mold allergy [7].

Fungal allergy in asthma (fungal asthma) is associated with increased disease severity and lung damage in adults and children [8,9]. Most fungi are either mesophilic (optimal growth occurring at 18 to 22 °C, and unable to grow at body temperature) or thermophilic (can grow at body temperature but are unable to grow below 20 °C, so are not present in the environment) and, thus, are rarely associated with human infections and allergies [10]. Thermotolerant fungi, especially *Aspergillus fumigatus*, grow equally well in the environment and at human body temperature and are associated with human disease, including fungal allergy. In increasing order of severity, the allergic diseases caused by *A. fumigatus* include fungal (aspergillus) sensitization, severe asthma with fungal sensitization (SAFS), and allergic bronchopulmonary aspergillosis (serologic, bronchiectasis, and with high-attenuation mucus) [11].

*Aspergillus* sensitization (AS) is considered the first step in the development of ABPA [12]. AS is currently diagnosed based on raised IgE against *A. fumigatus* antigens (≥0.35 kUA/L on the fluorescent enzyme immunoassay, Phadia platform) or immediate (type 1) cutaneous hyperreactivity on skin testing with *A. fumigatus* antigens, using either intradermal injection or prick test [11,13]. On the other hand, allergic bronchopulmonary aspergillosis (ABPA) is a complex immunological disorder caused by increased type-2 immune responses mounted against *A. fumigatus* colonizing the tracheobronchial tree of children with asthma or cystic fibrosis [14,15]. A related disorder is allergic bronchopulmonary mycosis (ABPM), an ABPA-like syndrome caused by fungi other than *A. fumigatus* [16]. Children with ABPA manifest with difficult-to-treat asthma, flitting and fleeting pulmonary opacities, expectoration of brownish-black mucus plugs, bronchiectasis, and lung collapse [17].

The Rosenberg–Patterson criteria have been the most widely used standard for diagnosing ABPA [18]. The Patterson criterion has eight major components (bronchial asthma, immediate cutaneous hypersensitivity to *A. fumigatus* antigen, serum total IgE levels ≥ 417 IU/mL, serum *A. fumigatus* specific IgG or IgE levels more than twice the mean values in patients with *Aspergillus*-sensitized asthma, central bronchiectasis on computed tomography [CT] of the chest, serum precipitins against *A. fumigatus*, fleeting or fixed pulmonary opacities on chest radiograph, and peripheral blood eosinophil count ≥ 1000 cells/mL) and three minor components (sputum cultures demonstrating the growth of *A. fumigatus*, expectoration of brownish-black mucus plugs, type III skin reactions to *A. fumigatus* antigen). There were several limitations to using the Patterson criteria. There was no consensus on the number of the components of the major and minor criteria required for the diagnosis, no agreement on the cut-off values of various immunological tests, and rendering equal weightage to all the individual components.

The International Society for Human and Animal Mycology (ISHAM) formed the “ABPA in asthmatics” working group to simplify the diagnosis and management of ABPA. The ISHAM ABPA working group has suggested more objective criteria for ABPA [19]. Based on these criteria, a diagnosis of ABPA can be made in the presence of a predisposing condition (asthma, cystic fibrosis, or others), when both the obligatory criteria (serum *A. fumigatus*-specific IgE levels > 0.35 kUA/L or positive type I *Aspergillus* skin test; and, serum total IgE levels > 1000 IU/mL) are met along with at least two of the three additional criteria (presence of precipitating or IgG antibodies against *A. fumigatus* in serum, thoracic imaging findings consistent with ABPA, and peripheral blood eosinophil count ≥ 500 cells/µL). The ISHAM-ABPA working group criteria were proposed in 2013. In the last decade, emerging evidence has indicated that *A. fumigatus*-IgE is better than skin tests [20,21], *A. fumigatus*-IgG estimation by enzyme immunoassays is superior to gel diffusion tests [22,23], CT performs better than chest radiographs [21], and the serum total IgE cut-off of 500 IU/mL is more sensitive than 1000 IU/mL [21]. We then suggested minor modifications to the ISHAM-ABPA working group criteria. The modified ISHAM-ABPA working group criteria include a predisposing condition (asthma, cystic fibrosis, or others), the presence of both the obligatory criteria (serum *A. fumigatus*-specific IgE levels > 0.35 kUA/L and serum total IgE levels > 500 IU/mL), and at least two of the three additional criteria (presence of IgG antibodies against *A. fumigatus* in serum, CT thorax showing bronchiectasis, and peripheral blood eosinophil count ≥ 500 cells/µL). By incorporating these changes, the group demonstrated better sensitivity to the modified (100%) than the original criteria (89%) [21]. The Japan ABPM Research Program criteria have proposed another criterion in asthmatic subjects with ABPM [24]. The criteria included ten components (asthma, peripheral blood eosinophil count ≥ 500 cells/µL, serum total IgE ≥ 417 IU/L, immediate cutaneous hypersensitivity or elevated serum fungi-specific IgE, presence of precipitins or specific IgG for filamentous fungi, filamentous fungus growth in sputum cultures or bronchial lavage fluid, presence of fungal hyphae in bronchial mucus plugs, central bronchiectasis on CT thorax, presence of mucus plugs in central bronchi based on CT or bronchoscopy or mucus plug expectoration history, and high-attenuation mucus on CT chest). Subjects fulfilling at least six of the ten components are labeled as definite ABPA/ABPM, while those meeting five are categorized as probable ABPA/ABPM. The Japanese criteria are more sensitive than the Rosenberg–Patterson and the ISHAM-working group criteria in asthmatic subjects with ABPM [24]. The limitations of the Japanese criteria include overreliance on microbiological criteria and the requirement of bronchoscopy. It is essential to note that the sputum or bronchial lavage fluid cultures have low sensitivity in the diagnosis of ABPA. Moreover, there is often a dissociation between colonizing and sensitizing fungi [25]. Notably, all the criteria mentioned above have been developed in the adult population, and there are no criteria specific to childhood ABPA [14].

While aspergillus-related allergy disorders are fairly well characterized in adults [19,26], the prevalence of AS and ABPA in children remains unknown [14]. It will be of interest to pediatricians if the prevalence of AS and ABPA is known in asthmatic children, as it can influence the implementation of screening programs. Herein, we systematically review the literature for estimating the prevalence of AS and ABPA in children with bronchial asthma.

## 2. Methods

We performed a systematic review of the literature to evaluate the prevalence of AS and ABPA among asthmatic children. The review protocol was registered at the PROSPERO database (crd.york.ac.uk; CRD42022352448). An Institute ethics committee approval was not necessary as this was a meta-analysis of published studies. The results are presented per the Meta-analysis of Observational Studies in Epidemiology (MOOSE) guidelines [27].

Search strategy: We (RA, VM) systematically queried the PubMed and Embase databases from inception, without language restrictions, to identify the relevant studies. We used the following search strategy: “allergic bronchopulmonary aspergillosis” OR “abpa” OR “allergic bronchopulmonary mycosis” OR “fungal sensitization” OR “fungal sensitisation” OR “fungal allerg*” OR “mould allerg*” OR “mold allerg*” OR “fungal asthma” OR “aspergillus sensitization” OR “aspergillus sensitisation” OR “aspergillus hypersensitivity” OR “mould sensitisation” OR “mould sensitization” OR “mold sensitisation” OR “mold sensitization” OR “mold sensitivity” OR “mould sensitivity” OR Aspergillosis, Allergic Bronchopulmonary [Mesh]. We further supplemented the search by reviewing our files and the references of the articles included.

Initial review of studies: We transferred the search results to a citation manager (Endnote X9.3.3, Clarivate Analytics, Philadelphia, PA, USA). Three authors (R.A., V.M., I.S.S.) screened the articles by title and abstract, retrieved the full text of each relevant citation, and assessed the articles for inclusion. Any disagreement was resolved by discussion between the authors.

Study selection: We included studies enrolling pediatric asthma (<18 years) irrespective of the study setting and reporting the prevalence of AS or ABPA in asthma. We defined aspergillus sensitization as either immediate cutaneous hyperreactivity to *A. fumigatus* antigens (either commercial or in-house) or raised *A. fumigatus*-specific IgE or both. For ABPA, we recorded the criteria used by the different authors. We excluded the following: (1) abstracts, editorials, review articles, and case reports; (2) studies describing the occurrence of AS or ABPA in subjects with cystic fibrosis; (3) studies conducted in the adult population; and (4) studies not providing the number of asthmatics screened.

Data extraction: We retrieved the following information from the included articles on a predefined data extraction form: (1) citation details, including the country where the research was conducted; (2) study type (prospective or retrospective); (3) age and sex of children with asthma and ABPA; (4) duration and severity of asthma; (5) procedure for detecting AS (intradermal injection, skin prick test, or *A. fumigatus*-specific IgE), if the studies provided data using both skin test and IgE estimation, we used the higher prevalence reported by either of the methods; (6) commercial or in-house *A. fumigatus* antigens used for diagnosing AS; (7) details of the criteria used for diagnosing ABPA; and (8) radiological subtype of ABPA (serologic [no bronchiectasis], bronchiectasis, or high-attenuation mucus).

Quality assessment of the studies: We used the Newcastle–Ottawa Scale (NOS) to assess the quality (modified for cross-sectional studies) of the included studies [28,29]. The scale has three components, namely, study selection (score 0 to 5), comparability (score 0 to 2), and outcome (0 to 3). The overall NOS score can range from 0 to 10. We rated the study as fair and good when the score was <8 or ≥8, respectively.

Outcomes: The primary outcome was to analyze the prevalence of AS in asthmatic children. The secondary outcome was to evaluate the prevalence of ABPA in children with bronchial asthma.

Pooled effect: We computed the prevalence of AS and ABPA in each study by calculating the proportion (number of subjects with AS or ABPA divided by the total number of asthmatics) with 95% confidence interval(s) (CI) [30]. The proportions were first turned into a quantity (the Freeman–Tukey variant of the arcsine square root transformed proportion) suitable for the random effects model [31,32]. The pooled proportion was calculated as the back-transform of the weighted mean of the transformed proportions, using DerSimonian–Laird weights for the random effects model, in the presence of significant heterogeneity [33]. We used StatsDirect (version 3.3.5; StatsDirect, Ltd., Cheshire, UK (http://www.statsdirect.com, accessed on 1 March 2022)) to perform the meta-analysis.

Heterogeneity: We calculated the impact of heterogeneity on the pooled estimates of the individual outcomes (prevalence of AS and ABPA) using the Higgins inconsistency index (I^2^) [34]. The Higgins index measures the inconsistency between the study results, interpreted as the approximate proportion of total variation in the study estimates due to heterogeneity rather than sampling error. An I^2^ value ≥50 percent indicates significant heterogeneity.

Publication bias: We checked the presence of publication bias by constructing a funnel plot (proportion in the horizontal axis plotted against the standard error of proportion in the vertical axis). The proportion estimates from smaller studies usually are scattered above and below the summary estimate producing a triangular or funnel shape, indicating the absence of publication bias. We also checked for publication bias using three statistical tests: (1) The Egger test evaluates the asymmetry of the funnel plot. The Egger test is for Y intercept = 0 from a linear regression of normalized effect estimate (proportion divided by its standard error) against precision (reciprocal of the standard error of the proportion); [35] (2) The Harbord test is similar to the Egger test but uses a modified linear regression method to reduce the false-positive rate; [36] and, (3) The Begg and Mazumdar test analyzes the interdependence of variance and proportion with a rank correlation method [37].

Subgroup analysis: We performed a subgroup analysis to evaluate the factors influencing heterogeneity. We included the following variables: (1) study design (prospective versus retrospective); (2) the country from where the study was published (India vs. the rest of the world); and (3) the type of economy of the country (developed vs. developing) [38].

## 3. Results

Search results: We searched the databases and obtained 11,695 records. After discarding duplicates, we screened 9031 records and excluded 8942 articles and 62 abstracts. Finally, we reviewed the full text of 27 articles and included 16 for the current review (Figure 1) [9,39,40,41,42,43,44,45,46,47,48,49,50,51,52,53].

Study characteristics: The 16 studies included 2468 asthmatic children (Table 1). All studies were conducted at tertiary care centers except one that was population-based (and reported only AS) [45]. Ten studies were prospective, while six were retrospective. Most studies included a mixed asthma population, while five were conducted on moderate-to-severe asthmatic children [9,40,46,47,53]. Four studies each were published from India [40,47,51,53] and the United States [39,43,45,46], and two from Finland [41,42]. One study each was published from Iran [44], Saudi Arabia [48], South Korea [49], Germany [50], Egypt [52], and the United Kingdom [9].

Study quality: The median (interquartile range) NOS score was 9 (8 to 10). We classified all the studies as good (Appendix A). The only issues were related to a small sample in most studies, the lack of inclusion of children with intermittent asthma, and lower numbers of mild asthmatic children.

Primary outcome (prevalence of AS in asthma): A total of 15 studies (2361 subjects) reported the prevalence of AS in asthmatic children. Five studies used skin prick tests, four used intradermal testing, five used *A. fumigatus*-specific IgE estimation, and one used both SPT and *A. fumigatus*-specific IgE (Table 1). Twelve studies used commercial *A. fumigatus* antigens, two used in-house antigens, while data for the antigen were unavailable from one study. The prevalence of AS in asthma (Figure 2) ranged from 0 to 61.3%, with a pooled prevalence of 16.1% (95% CI, 9.3 to 24.3).

Heterogeneity and publication bias: There was significant heterogeneity for the primary outcome, with an I^2^ value of 95.9%. The funnel plots showed evidence of publication bias (Figure 3). The statistical tests also indicated publication bias by two methods (Begg–Mazumdar: Kendall’s tau = 0.524, *p* = 0.006; Egger: bias = 6.24, *p* = 0.002; Harbord–Egger: bias = 4.22, *p* = 0.32).

Subgroup analysis: We performed univariate analysis to explore the causes of heterogeneity (Table 2). The prevalence of AS was significantly higher in the following study groups: prospective studies (rather than retrospective), studies published from India (compared to the rest of the world), and studies published from developing countries (vs. developed countries). However, there was significant statistical heterogeneity in all the subgroups also.

Secondary outcome: A total of 5 studies that included 505 asthmatic children provided data to calculate the prevalence of ABPA [9,40,46,47,53]. Three studies used the Patterson criteria (or its modification) [18], while two used the ISHAM-ABPA working group criteria [19]. The diagnosis of ABPA with the Patterson criteria was made variably using five to eight components. The prevalence of ABPA in asthma varied from 0 to 32.7% (Figure 4), with a pooled prevalence of 9.9% (95% CI, 0.81 to 27.6). Four studies provided data for calculating the prevalence of ABPA in aspergillus-sensitized asthma [9,46,47,53]. The pooled prevalence of ABPA in aspergillus-sensitized asthma was 20.5% (95% CI, 0 to 65.7).

Heterogeneity and publication bias: There was significant heterogeneity for the secondary outcome, with an I^2^ value of 96.8%. The funnel plot (Appendix A) suggested publication bias. The statistical tests, however, did not reveal evidence of publication bias (Begg–Mazumdar: Kendall’s tau = 1, *p* = 0.08; Egger: bias = 22.01, *p* = 0.14; Harbord–Egger: bias = 1.47, *p* = 0.93).

## 4. Discussion

We found a high prevalence of AS (16%) and ABPA (10%) in children with bronchial asthma. Further, there was a high prevalence of ABPA in asthmatic children with AS (21% in the current study). The prevalence of AS was significantly higher in prospective studies, studies from India or developing countries. The current study is the first to systematically review the global prevalence of AS and ABPA in asthmatic children.

Despite being the most common respiratory disorder in children, pediatric asthma remains poorly understood. The differential diagnosis of asthma, particularly severe asthma, is broader in children than adults and includes congenital structural abnormalities (tracheomalacia, webs, and others), foreign body aspiration, primary ciliary dyskinesia, immunodeficiency diseases, cystic fibrosis, recurrent aspiration, and others. The evaluation of severe asthma is further hampered by difficulty establishing a diagnosis, excluding comorbidities and alternate diagnosis, and non-adherence to inhaled medications (especially in adolescents) [54]. Identifying distinct phenotypes and using various biomarkers usually guides the management of severe pediatric asthma [5]. While the phenotypes may suggest the pattern of ongoing airway inflammation and influence the choice of various therapies (including biologicals for asthma), identifying the triggers for the underlying inflammation is also essential [5,55].

Of the various triggers, thermotolerant molds assume greater significance as the mold is actively growing the airways and thus is amenable to treatment with antifungal agents. In children, antifungal agents are safer than prednisolone and less expensive than biological agents. Indeed, home environment assessment is considered an essential component of evaluating children with severe asthma, and molds are common indoor allergens [56]. While fungal sensitization has been recognized as an important contributor to the severity of asthma in children [9], their true burden remains unknown since routine evaluation is not performed [57].

The high prevalence of AS in the current study suggests that fungal allergy, particularly to *A. fumigatus*, should be actively sought, at least in children with persistent asthma. We found the prevalence of AS to range from 0 to 61%. We can speculate several causes for the wide range in prevalence. The prevalence of AS was higher in prospective than retrospective studies, understandably due to the limitations of retrospective analysis. The prevalence of AS was higher in studies published in India (and developing countries), possibly due to environmental and genetic factors [58]. There is a high environmental burden of *A. fumigatus* spores in rural and urban areas in India [59]. Ethnic (genetic) differences also play an important role in determining allergic sensitization [60]. The other important reason for the wide prevalence is the varying time points for screening asthmatic children for AS in the studies. Notably, many asthmatic children can develop AS later despite being unsensitized at an earlier point in time. In two different studies, it was shown that in asthmatic patients without AS, de novo sensitization could develop later, over the next one to two decades [61,62].

We noted that the prevalence of ABPA also varied widely (0–33%) in the present study. Of the five studies included in the meta-analysis, two (one each from the United Kingdom and the United States) did not report any case of ABPA [9,46]. The other three studies were from India and reported a high prevalence of ABPA, ranging from 11 to 33 percent [40,47,53]. Our data suggest a high burden of AS and ABPA, particularly in India. India has approximately 444 million children, and an estimated 7.9% have asthma, i.e., 35 million [63]. From our study, the estimated burden of AS and ABPA among asthmatic children in India alone would be 11 million and eight million, respectively. The variation in prevalence between India and other countries can be attributed to environmental and genetic factors. However, another important possibility must be considered, particularly in countries where cystic fibrosis is uncommon [64,65]. A recent study from adult asthmatics found a higher frequency of single nucleotide polymorphisms (SNPs) in the CFTR gene among ABPA, complicating asthma compared to asthmatics and healthy controls [66]. ABPA is also regarded as a cystic fibrosis transmembrane conductance regulator (CFTR)-related disease [67,68], and the latter may be more relevant in children than adults. In a non-Caucasian population with a low CF prevalence and lack of sweat chloride testing (or genetic analysis), some CF variants with ABPA may be classified as ABPA, complicating asthma. Additionally, the lack of uniform criteria for diagnosing pediatric ABPA could also explain the differences in prevalence [14].

ABPA is considered rare in children with asthma [6]. However, the rarity of ABPA in children seems implausible given the high prevalence of AS in children (16% in the current analysis). A significant contributor could be the lack of routine screening of all asthmatic children for ABPA. ABPA is generally considered to complicate the course of difficult-to-treat or uncontrolled asthma. Interestingly, in a series consisting of 155 cases of adult ABPA, almost 19% of the patients had well-controlled asthma [69]. The occurrence of ABPA in controlled asthma underscores the need for investigating all asthmatic children for ABPA, irrespective of asthma control, especially in tertiary care settings. The other cause for the lower prevalence of ABPA in children could be the overlapping criteria for diagnosing SAFS and serological ABPA [17]. For instance, raised serum total IgE or raised *A. fumigatus*-specific IgG are considered components of SAFS [6], while they are dominant components of ABPA in adults. Several children classified as AS in pediatric studies are likely ABPA. Moreover, a chest radiograph is insensitive in identifying bronchiectasis [70], and the less frequent performance of CT scans in children than in adults with severe asthma could also result in the misclassification of ABPA as SAFS.

What are the clinical implications of the current study? Do all asthmatic children require evaluation for AS? The answer is not known. Wheezing is a common problem in children, and the prevalence of wheezing varies across countries and even within the country [71]. Variation of asthma symptoms with age also poses difficulty in assigning severity to childhood asthma. Hence, periodic assessment is required, and subjects with persisting wheeze may be considered for further evaluation, including routine assessment for AS. More data and systematic evaluation are required before definite recommendations for screening all asthmatic children for AS. Nevertheless, screening for AS should be strongly considered in tertiary care settings, and in children with persistent asthma because the presence of AS in asthmatic children, even without ABPA, is associated with poor lung function and recurrent hospital admissions [9,46,51]. Itraconazole improved the quality of life in adult patients with SAFS [72]. Similar studies are required in pediatric SAFS. In pediatric practice, performing specific IgE against allergens is usually guided by environmental history and assessment [55]. However, a detailed indoor (home, school, and other places) and outdoor environment assessment may not be practical. Hence, evaluation for allergen sensitization that is common and of greater significance should be considered for screening among asthmatic children. Further, the lack of a consensus definition for SAFS makes comparing studies difficult [6]. In the present review, most studies using the Patterson criteria for diagnosing ABPA used components varying from five to eight that can affect the diagnostic performance [20]. We suggest that future studies follow a uniform method and use *A. fumigatus*-specific IgE and skin test (rather than skin test alone) for detecting AS and the recently proposed modified ISHAM-ABPA working group criteria for diagnosing ABPA [21,73]. Not all asthmatic children residing in the same environment develop AS. Thus, the genetic predisposition for AS needs to be explored in future studies. There is also a need to develop criteria specific to pediatric ABPA. Notably, ABPA is considered a treatable trait in bronchiectasis [74,75]. Early recognition and treatment of ABPA are essential to prevent progressive lung damage. Our study results suggest that AS and ABPA meet the requirement for implementing a screening program in tertiary care clinics for early detection [76].

Our study has a few limitations. The major limitation is the small number of studies reporting the prevalence of AS and ABPA. The other limitation is the methodological (varying immunological tests and diagnostic criteria) and statistical heterogeneity. We used the random effects model to partially offset the statistical heterogeneity and performed subgroup analysis to explore the causes of heterogeneity. The variation in the prevalence of AS across the included studies could also be due to the differences in antigen and the method used for testing, apart from the severity of the asthmatic children enrolled. The duration of asthma is another crucial factor in determining the development of AS. Unfortunately, we did not have sufficient data to assess the relationship between the duration of asthma and the prevalence of AS or ABPA. Longitudinal cohorts performing serial monitoring for AS among asthmatic children may improve our understanding and guide us on the optimal timing for screening. While we have reported data on the prevalence of AS, we do not know what proportion of AS subjects have severe asthma. Most studies included in our review did not provide AS data separately for severe and non-severe asthmatics. Similar to adult data [77], most studies included in our meta-analysis were from referral centers, and the prevalence of ABPA does not represent the population prevalence. The strengths of our study include an exhaustive literature search and predefined criteria for study assessment and statistical analysis.

In conclusion, we found a high prevalence of *Aspergillus* sensitization and ABPA in children with bronchial asthma. Due to the paucity of studies in children, more data are required from different ethnic groups using a uniform methodology to ascertain the true prevalence of AS and ABPA in asthmatic children.

## Figures and Tables

**Figure 1 diagnostics-13-00922-f001:**
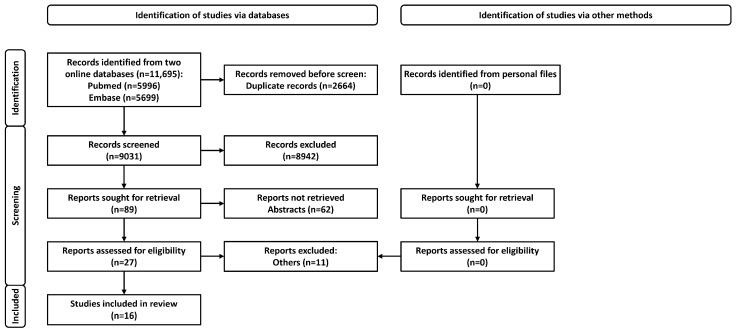
Study selection process for the systematic review.

**Figure 2 diagnostics-13-00922-f002:**
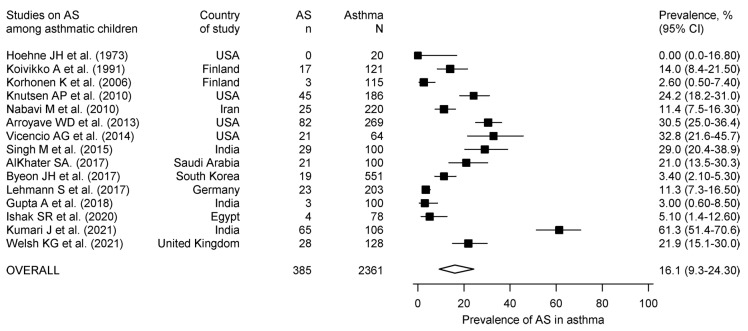
Forest plot for the pooled prevalence of aspergillus sensitization in asthmatic children. The prevalence reported in the individual studies is represented by the black square with horizontal bars indicating the 95% confidence interval. The diamond at the bottom refers to the pooled prevalence with a 95% confidence interval [9,39,41,42,43,44,45,46,47,48,49,50,51,52,53].

**Figure 3 diagnostics-13-00922-f003:**
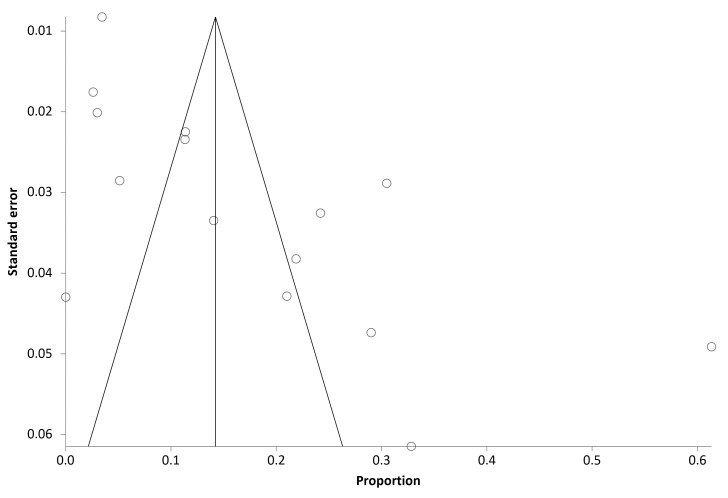
Funnel plot showing significant publication bias for the outcome of aspergillus sensitization in children with bronchial asthma. The proportion and standard error are displayed along the horizontal and vertical axes, respectively.

**Figure 4 diagnostics-13-00922-f004:**
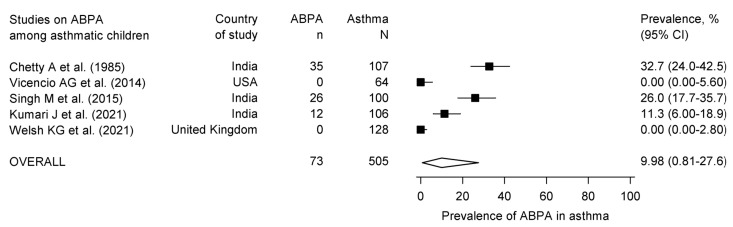
Forest plot showing the pooled prevalence of allergic bronchopulmonary aspergillosis in pediatric asthma. The prevalence in individual studies is shown by the black square with horizontal bars indicating the 95% confidence interval. The diamond denotes the pooled prevalence with a 95% confidence interval [9,40,46,47,53].

**Table 1 diagnostics-13-00922-t001:** Studies describing the prevalence of Aspergillus sensitization (AS) or allergic bronchopulmonary aspergillosis (ABPA) in children with bronchial asthma.

Study	State	Type of Study	No. of Children with Bronchial Asthma	Asthma Severity	Age, Years	Sex M/F	Type of Skin Test	Type of Antigen	Criteria Used for Diagnosis of ABPA	ABPA S/B/HAM, *n*	Prevalence of AS in Asthma (*n*/N)	Prevalence of ABPA in Asthma (*n*/N)	Prevalence of ABPA in AS (*n*/N)
Hoehne JH, et al. (1973) [39]	United States	Retrospective	20	Varying	-	-	Intradermal	In-house	-	-	0/20	-	-
Chetty A, et al. (1985) [40]	India	Retrospective	107	Varying but perennial asthma	Children	ABPA: 24/11	Intradermal	Commercial (Curewell lab, India)	Major: A/R/T/E/P/I/Minor: S	-	-	35/107	-
Koivikko A, et al. (1991) [41]	Finland	Retrospective	121	Varying	Asthma: 1–15	-	*A. fumigatus*-IgE	Commercial (Phadebas-RAST)	-	-	17/121	-	-
Korhonen K, et al. (2006) [42]	Finland	Prospective	122	Varying	Asthma: 1–6	-	SPT	Commercial (ALK)	-	-	3/115	-	-
Knutsen AP, et al. (2010) [43]	United States	Prospective	186	Varying	Asthma: 10.5 ± 4.0 y	Asthma: 103/83	SPT	Commercial (Multi-test II, Lincoln diagnostics, USA)	-	-	45/186	-	-
Nabavi M, et al. (2010) [44]	Iran	Prospective	220	Varying	<18 years	-	SPT	-	-	-	25/220	-	-
Arroyave WD, et al. (2013) [45]	United States	Prospective	351	Varying	Asthma: 1–17	Asthma: 197/154	*A. fumigatus*-IgE	Commercial (Pharmacia Diagnostics, ImmunoCAP 1000, USA)	-	-	82/269	-	-
Vicencio AG, et al. (2014) [46]	United States	Prospective	64	Moderate-severe persistent asthma	Asthma: mean (range), 9 (6–12)	-	*A. fumigatus*-IgE	RAST panel	Major: A/R/T/E/P/	-	21/64	0/64	0/21
Singh M, et al. (2015) [47]	India	Prospective	100	Poorly controlled	Asthma: 9.6 ± 2.2	Asthma: 75/25	Intradermal	In-house	Major: A/R/T/E/P/I/C/SMinor: C/S/B [number not mentioned]	5/21/0	29/100	26/100	26/29
AlKhater SA, et al. (2017) [48]	Saudi Arabia	Retrospective	100	Varying	Asthma: 9.0 ± 2.9	Asthma: 68/32	SPT	Commercial (Stallergenes, France)	-	-	21/100	-	
Byeon JH, et al. (2017) [49]	South Korea	Retrospective	551	Varying	Asthma: 9.1 ± 2.5 y	Asthma: 345/206	SPT	Commercial	-	-	19/551	-	-
Lehmann S, et al. (2017) [50]	Germany	Retrospective	207	Varying	Asthma: 8.6 ± 3.5	Asthma: 136/71	*A. fumigatus*-IgE	Commercial (UniCAP, Phadia, Thermo Fisher, Sweden)	-	-	23/203	-	-
Gupta A, et al. (2018) [51]	India	Prospective	100	Varying	Asthma: 10.3 (mean)	Asthma: 79/21	Intradermal	In-house	-	-	3/100	-	-
Ishak SR, et al. (2020) [52]	Egypt	Prospective	78	Varying	<18 years	-	*A. fumigatus*-IgE	Commercial (ImmunoLINE IgE Perennial, IMMUNOLAB, Germany)	-	-	4/78	-	-
Kumari J, et al. (2020) [53]	India	Prospective	235	Poorly controlled	Asthma: 10.2 ± 2.6	Asthma: 72/34	SPT or *A. fumigatus*-IgE	Commercial (Allergopharma, Merck, Reinbeck, Germany); ImmunoCAP, Phadia	ISHAM-AWG	7/5/0	65/106	12/106	12/65
Welsh KG, et al. (2021) [9]	United Kingdom	Prospective	128	Varying (acute asthma [mild-moderate: 33/39; severe:6/39],chronic asthma [mild-moderate: 51/99; severe: 48/99])	Median (range)Acute asthma: 9 (5–15); Chronic asthma: 11 (5–17)	Acute: 24/15Chronic: 57/42	SPT	Commercial (Soluprick, ALK-Abello, Hørsholm, Denmark)	ISHAM-AWG	0/0/0	Acute: 7/35.Chronic: 21/93Total: 28/128	0/128	0/28

FEIA: fluorescent enzyme immunoassay; RAST: radioallergosorbent test; SPT: skin prick test. **Criteria for ABPA**: Major (A—asthma, R—radiologic opacities, T—immediate positive skin test, E—eosinophilia, P—precipitins to *A. fumigatus*, I—IgE elevated, C—central bronchiectasis, S—specific IgG/IgE to *A. fumigatus*); Minor (C—sputum cultures of *A. fumigatus*, S—type III skin test positivity, B—brownish black mucus plugs). **International society for human and animal mycology—ABPA working group (ISHAM-AWG) criteria**: presence of all the following: asthma, positive type I skin reaction to *A. fumigatus* or *A. fumigatus*-specific IgE > 0.35 KUA/L, serum total IgE levels >1000 IU/mL; and ≥2 of the following: positive *A. fumigatus* precipitins or *A. fumigatus*-specific IgG >27 mgA/L, chest radiograph favoring ABPA, total eosinophil count >500 cells/µL. **Modified ISHAM-AWG criteria**: presence of all the following: asthma, *A. fumigatus*-specific IgE > 0.35 KUA/L, serum total IgE levels > 500 IU/mL; and ≥2 of the following: *A. fumigatus*-specific IgG > 27 mgA/L, computed tomography favoring ABPA, total eosinophil count > 500 cells/µL.

**Table 2 diagnostics-13-00922-t002:** Prevalence of aspergillus sensitization (AS) in various subgroups of asthmatic children.

	Number of Studies	Number of Subjects (AS/Asthma)	I^2^ Value	*p*-Value
All studies	15	385/2361	95.9%	
SUBGROUP ANALYSIS				
Study design				<0.0001
Prospective	10	305/1366 (22.3%)	95.6%	
Retrospective	5	80/995 (8.0%)	91.2%	
Country of publication				<0.0001
India	3	97/306 (31.7%)	98.1%	
Rest of the world	12	288/2055 (14.0%)	94.3%	
Developed vs. developing countries				<0.0001
Developed	9	238/1657 (14.4%)	94.3%	
Developing	6	147/704 (20.9%)	96.5%	

## Data Availability

All data are presented in the manuscript.

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
