# Peer review of "Aspergillus Sensitization and Allergic Bronchopulmonary Aspergillosis in Asthmatic Children: A Systematic Review and Meta-Analysis"

_diagnostics, 2023, doi:10.3390/diagnostics13050922_

Round 1
Reviewer 1 Report
Manuscript: Aspergillus sensitization and allergic bronchopulmonary aspergillosis in asthmatic children: a systematic review and metaanalysis
Systematic Review
The Authors in the systematic review and meta-analysis analysed current data on Aspergillus sensitization and allergic bronchopulmonary aspergillosis in children with asthma.
This is an interesting and well-written manuscript with a well-designed graphic design. Congratulation.
Author Response
Dear reviewers
Thank you very much for your kind comments. The manuscript has immensely benefited from these comments
Reviewer 1
Manuscript: Aspergillus sensitization and allergic bronchopulmonary aspergillosis in asthmatic children: a systematic review and metaanalysis
Systematic Review
The Authors in the systematic review and meta-analysis analysed current data on Aspergillus sensitization and allergic bronchopulmonary aspergillosis in children with asthma.
This is an interesting and well-written manuscript with a well-designed graphic design. Congratulation.
Response: A native English speaker has reviewed the manuscript to improve the language.
Reviewer 2 Report
Comments/Suggestions:
Agarwal and colleagues penned a systematic review and performed a meta-analysis on Aspergillus sensitization and allergic bronchopulmonary aspergillosis in asthmatic children.
They found a high prevalence of AS and ABPA in asthmatic children and suggested there is a need for community-based studies from different ethnicities using a standard methodology to ascertain the true prevalence of AS and ABPA in pediatric asthma.
· The author must be careful of the manuscript's spelling and other grammatical errors.
· The data size is not sufficient to reach any strong conclusion.
· Lack of statistical power.
· More literature surveys are required.
· The discussion part should be stronger.
· The conclusion part should be more specific.
· The figure and table quality should be improved.
Author Response
Dear reviewer
Thank you very much for your kind comments. The manuscript has immensely benefited from these comments.
Reviewer 2
Agarwal and colleagues penned a systematic review and performed a meta-analysis on Aspergillus sensitization and allergic bronchopulmonary aspergillosis in asthmatic children.
They found a high prevalence of AS and ABPA in asthmatic children and suggested there is a need for community-based studies from different ethnicities using a standard methodology to ascertain the true prevalence of AS and ABPA in pediatric asthma.
- The author must be careful of the manuscript's spelling and other grammatical errors.
Response: A native English speaker has reviewed the manuscript to improve the language.
- The data size is not sufficient to reach any strong conclusion.
- Lack of statistical power.
- More literature surveys are required.
Response: we agree with the reviewer that there is paucity of data in pediatric literature on Aspergillus sensitization and ABPA. We have included this point as a limitation, including the requirement of more surveys.[lines 74-77, 379-80, 396-99]
- The discussion part should be stronger.
Response: we agree with the reviewer, and we have rewritten the discussion part.
- The conclusion part should be more specific.
Response: the conclusion portion has been made more specific [lines 74-77, 396-99]
- The figure and table quality should be improved.
Response: the figure and tables have been made better
Round 2
Reviewer 2 Report
Recommendation for publication